# Absence of localization in interacting spin chains with a discrete symmetry

Benedikt Kloss [1] ✉, Jad C. Halimeh[2,3], Achilleas Lazarides[4] &
Yevgeny Bar Lev [5]

Novel paradigms of strong ergodicity breaking have recently attracted significant attention in condensed matter physics. Understanding the exact conditions required for their emergence or breakdown not only sheds more light on thermalization and its absence in closed quantum many-body systems, but it also has potential benefits for applications in quantum information technology. A case of particular interest is many-body localization whose conditions are not yet fully settled. Here, we prove that spin chains symmetric under a combination of mirror and spin-flip symmetries and with a non-degenerate spectrum show finite spin transport at zero total magnetization and infinite temperature. We demonstrate this numerically using two prominent examples: the Stark many-body localization system (Stark-MBL) and the symmetrized many-body localization system (symmetrized–MBL). We provide evidence of delocalization at all energy densities and show that delocalization persists when the symmetry is broken. We use our results to construct two localized systems which, when coupled, delocalize each other. Our work demonstrates the dramatic effect symmetries can have on disordered systems, proves that the existence of exact resonances is not a sufficient condition for delocalization, and opens the door to generalization to higher spatial dimensions and different conservation laws.

One of the basic assumptions of classical or quantum statistical mechanics is that interacting many-body systems thermalize, approaching local thermodynamic equilibrium under unitary dynamics. This assumption is not satisfied for localized systems, in which transport is arrested. Two well-known examples are strongly disordered "many-body localized" (MBL) systems[1–5], and clean systems with a strong tilted potential ("Stark-MBL")[6,7]. Significant suppression of dynamics was experimentally observed in both MBL[8,9] and Stark-MBL systems[10,11].

What are the main mechanisms destabilizing such localized phases? In noninteracting systems, resonances–distinct regions in space with close energies of the single-particle orbitals[12]–are the natural cause of instability towards delocalization. Lowering the disorder strength increases their density, eventually leading to delocalization at three or higher dimensions[12]. The resonances also give the dominant contribution to ac conductivity in localized systems[13–17] and can induce a non-local response. For interacting systems and under the assumption that the levels of the many-body spectrum do not attract each other, it has been rigorously shown that many-body resonances (which must now take into account the local interaction energy) cannot delocalize one-dimensional disordered systems[18,19]. The proof does not apply for higher dimensions, and it is

[1]Center for Computational Quantum Physics, Flatiron Institute, 162 Fifth Ave, New York, NY 10010, USA. [2]Department of Physics and Arnold Sommerfeld Center for Theoretical Physics (ASC), Ludwig-Maximilians-Universität München, Theresienstraße 37, D-80333 München, Germany. [3]Munich Center for Quantum Science and Technology (MCQST), Schellingstraße 4, D-80799 München, Germany. [4]Interdisciplinary Centre for Mathematical Modelling and Department of Mathematical Sciences, Loughborough University, Loughborough, Leicestershire LE11 3TU, UK. [5]Department of Physics, Ben-Gurion University of the Negev, Beer-Sheva 84105, Israel. ✉e-mail: bene.kloss@gmail.com

currently unclear if localization is possible for two and higher-dimensional interacting systems[20,21].

Systems, where the resonances are caused by symmetries, have attracted special attention. In these systems, the rigorous proof of localization does not apply[18,19]. The common lore is that discrete compact symmetries do not affect localization[22–26], however translation-invariant glassy systems[7,27] and also systems with continuous non-Abelian symmetries can become delocalized due to the proliferation of resonances[28,29]. Symmetry-assisted delocalization is however not stable to the addition of symmetry-breaking perturbations that lift many of the exact resonances[30].

A number of studies argue that MBL is unstable to the existence of delocalized inclusions, ruling out the existence of a mobility edge[31], and even the MBL transition itself[32–34]. This delocalization mechanism was numerically explored by embedding of thermal regions in MBL systems[21,35–37], or by coupling the system to a Markovian bath[38,39]. It is not clear if a similar mechanism is present in clean localized systems such as Stark-MBL. More recent numerical studies argue that a bona fide many-body localization does not exist[37,38,40–42], but there is a glassy phase with possibly logarithmic growth of number entropy[43,44]. Other studies argue that observed delocalization is mostly a manifestation of the limitations of numerical studies[45–49].

As evidenced by the preceding discussion, the landscape is not yet clearly mapped, and this to a large extend is because numerical and approximate analytical studies have trouble distinguishing between slow and genuinely localized dynamics. It is therefore imperative to have rigorous results, and this is what motivates our work.

In this article, we prove that localization is absent in a large class of many-body spin systems with a non-degenerate spectrum. This class consists of all systems symmetric under the combination of spatial mirroring and spin flipping. By numerically verifying that the non-degeneracy assumption is fulfilled for interacting Stark-MBL and appropriately symmetrized disordered problems, we thus rule out localization in these systems and then explore the stability of these results to symmetry-breaking perturbations. Finally, we utilize our result to construct two localized systems that delocalize each other (see Fig. 1 for an illustration).

## Results

### General argument

We consider a spin chain of length $L$ described by a Hamiltonian $\hat{H}$, and assume the following:

**Assumption 1.** $\hat{H}$ has a non-degenerate spectrum.

**Assumption 2.** Total magnetization is conserved, $\left[\hat{H}, \sum_i \hat{S}_i^z\right] = 0$.

**Assumption 3.** The Hamiltonian is symmetric under a combination of a mirror symmetry and a spin-flip symmetry defined as

$$\begin{aligned} \hat{P}\hat{S}_i^z\hat{P} &= -\hat{S}_{L-i+1}^z, \\ \hat{P}\hat{S}_i^\pm\hat{P} &= \hat{S}_{L-i+1}^\mp, \end{aligned} \quad (1)$$

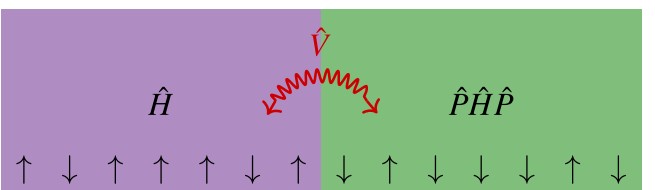

**Fig. 1 | Illustration of a symmetrized Hamiltonian.** Here $\hat{H}$ conserves the total magnetization and $\hat{V}$ represents the coupling between $\hat{H}$ and $\hat{P}\hat{H}\hat{P}$, such that $\hat{P}\hat{V}\hat{P} = \hat{V}$. The symmetry generator $\hat{P}$ mirrors and flips the spin-pattern on the left.

where $\hat{S}_i^z$ are spin operators of arbitrary spin size at site $i$, and $\hat{S}_i^\pm$ are their corresponding raising (lowering) operators.

Since $\hat{P}^2 = \hat{\mathbb{1}}$ its eigenvalues are ±1. The commutator $\left[\hat{P}, \sum_i \hat{S}_i^z\right] \neq 0$ unless the total magnetization vanishes, and therefore we project the Hamiltonian onto the zero total magnetization sector, namely, we work in the zero-magnetization sector.

We study spin transport by creating a spin excitation at site $j$ on top of some equilibrium state $\hat{\rho}$, such that $\left[\hat{\rho}, \hat{H}\right] = 0$, and assess its spreading using the connected spin-spin correlation function,

$$G_{ij}^\rho(t) = \left\langle \hat{S}_i^z(t)\hat{S}_j^z \right\rangle - \left\langle \hat{S}_i^z \right\rangle \left\langle \hat{S}_j^z \right\rangle, \quad (2)$$

where $\left\langle \hat{O} \right\rangle \equiv \mathrm{Tr}\left(\hat{\rho}\hat{O}\right)$. Taking the infinite-time average, $\overline{G_{ij}^\rho} = \lim_{T\to\infty} \frac{1}{T}\int_0^T d\bar{t}\, G_{ij}^\rho(\bar{t})$, and using Assumption 1 we obtain,

$$\overline{G_{ij}^\rho} = \sum_\alpha p_\alpha \langle \alpha | \hat{S}_i^z | \alpha \rangle \langle \alpha | \hat{S}_j^z | \alpha \rangle - \left\langle \hat{S}_i^z \right\rangle \left\langle \hat{S}_j^z \right\rangle, \quad (3)$$

where $0 \leq p_\alpha \leq 1$ are the eigenvalues of $\hat{\rho}$. For systems without spin transport, the spin excitation is expected to be localized in the vicinity of site $j$ at infinite times, $\overline{G_{ij}^\rho} - G_{ij}^\rho(t=0) \sim \exp\left[-|i-j|/\xi\right]$, where $\xi$ is the localization length. Note that, since $\hat{H}$ is symmetric with respect to $\hat{P}$, we have that $\overline{G_{ij}^\rho} = -\overline{G_{\check{i}j}^\rho}$ where $\check{i} = L - i + 1$ is the mirrored coordinate. Therefore, localization of the excitation around $j$ implies also localization around $\check{j}$, which can be arbitrarily distant from $j$. This however does not mean that the system is delocalized. While the spin excitation can move for arbitrary distances $|j - \check{j}|$, this is similar to resonant transfer between site $j$ and site $\check{j}$, which leaves the rest of the system localized. There is no transport in general. A similar situation occurs in the Anderson insulator[13] and MBL systems[50]. For systems that relax to equilibrium $\overline{G_{ij}^\rho} \to 0$ such that the excitation is uniformly spread over the lattice. Here the process is inherently many-body since it is not present for systems which can be mapped to noninteracting fermions, see Supplementary Note 1. To quantify the spreading of the excitation we use the mean-squared displacement (MSD),

$$\sigma_\rho^2(t) = \sum_{i=1}^L (i-j)^2 \left[G_{ij}^\rho(t) - G_{ij}^\rho(0)\right], \quad (4)$$

and its corresponding infinite-time average $\overline{\sigma_\rho^2}$. For delocalized states, the infinite-time averaged MSD scales as $\overline{\sigma_\rho^2} \sim L^2$, while for localized states $\overline{\sigma_\rho^2} \sim \xi^2$. We now prove that for systems satisfying the assumptions above, $\overline{\sigma_\rho^2} \sim L^2$, implying that at least a finite fraction of eigenstates are delocalized. For brevity, we only provide the sketch of the proof here; see Supplementary Note 3 for details.

We take $\hat{\rho} = \hat{\mathbb{1}}/\mathcal{N}$ where $\mathcal{N} = \binom{L}{L/2}$ is the Hilbert space dimension. This corresponds to setting $p_\alpha = 1/\mathcal{N}$ in Eq. (3), such that the infinite-time average of (4) becomes,

$$\begin{aligned} \overline{\sigma_\infty^2} &= \frac{1}{\mathcal{N}} \sum_{i=1}^L (i-j)^2 \sum_\alpha \langle \alpha | \hat{S}_i^z | \alpha \rangle \langle \alpha | \hat{S}_j^z | \alpha \rangle \\ &\quad - \frac{1}{\mathcal{N}} \sum_{i=1}^L (i-j)^2 \sum_\alpha \langle \alpha | \hat{S}_i^z \hat{S}_j^z | \alpha \rangle. \end{aligned} \quad (5)$$

We first note that $\mathcal{N}^{-1} \sum_\alpha \langle \alpha | \hat{S}_i^z \hat{S}_j^z | \alpha \rangle = \frac{1}{4(L-1)}$ and therefore the second term in (5) is $O(L^2)$, see Supplementary Note 3. To bound the first term we use the symmetry $\hat{P}$ and the identity,

$$\sum_{i=1}^L (i-j)^2 \langle \alpha | \hat{S}_i^z | \alpha \rangle = \left(\check{j} - j\right) \langle \alpha | \hat{D} | \alpha \rangle, \quad (6)$$

where $\hat{D} = \sum_i i \hat{S}_i^z$ is the dipole operator and $\check{i} = L - i + 1$ the mirrored coordinate. Inserting this identity into the first term in (5) and using a

combination of triangle and Hölder inequalities, we bound

$$\frac{1}{N} \left| \sum_{i=1}^{L} (i-j)^2 \sum_{\alpha} \langle \alpha | \hat{S}_i^z | \alpha \rangle \langle \alpha | \hat{S}_j^z | \alpha \rangle \right| \le \frac{|\bar{j}-j|}{2} \left( \frac{1}{N} \text{Tr} \hat{D}^2 \right)^{1/2}. \quad (7)$$

Since the second term in (5) can be exactly evaluated and scales as $L^2$ and it can be shown that $\langle \hat{D}^2 \rangle^{1/2} = O(L^{3/2})$, then for all $|j-\bar{j}| < O(L^{1/2})$ the second term is dominating in the thermodynamic limit which yields $\overline{\sigma_\infty^2} \sim L^2$. It is important to note that this is not an upper bound on $\overline{\sigma_\infty^2}$ but an asymptotic result, which implies delocalization of a finite fraction of eigenstates. Ideally we would like to prove that $\overline{\sigma_\infty^2} \sim L^2$ for all excitation sites $1 \le j \le L$ and not only sites found at distance $O(L^{1/2})$ from the center of the lattice. For the models we have considered numerically delocalization holds for all $j$.

The proof does not rule out localization in noninteracting Stark or Anderson problems, since due to the $\hat{P}$ symmetry there are degeneracies in the many-body spectrum invalidating Assumption 1, see also Supplementary Note 1.

In what follows, we numerically demonstrate that Assumption 1 is satisfied for two cornerstone models of localization in interacting systems and provide evidence of delocalization for all energy densities.

## Applications

We consider the Hamiltonian of a spin-1/2 chain of length $L$,

$$\hat{H} = \sum_{n=1}^{L-1} \left[ \frac{J}{2} \left( \hat{S}_n^+ \hat{S}_{n+1}^- + \hat{S}_n^- \hat{S}_{n+1}^+ \right) + \Delta \hat{S}_n^z \hat{S}_{n+1}^z \right] + \sum_{n=1}^{L} h_n \hat{S}_n^z, \quad (8)$$

where $\hat{S}_n^\pm$, $\hat{S}_n^z$ are spin-1/2 operators, $J$ is the strength of the flip-flop term, $\Delta$ is the strength of the Ising term and $h_n$ is an arbitrary magnetic field. For $h_n = -h_{\bar{n}}$ the Hamiltonian clearly satisfies Assumptions 2 and 3. In what follows we numerically verify that Assumption 1 is also satisfied for our choices of $h_n$. We consider two cases of ostensibly localized interacting systems: (a) $h_n = \gamma \left( n - \frac{L+1}{2} \right)$, such that all the single-particle states of the fermionic model are known to be localized for any $\gamma$ and for sufficiently large $\gamma$ the model is expected to be Stark many-body localized (Stark-MBL)[6,7]. (b) $h_n = -h_{\bar{n}}$, but otherwise randomly and uniformly distributed in the interval $[-W, W]$. We have verified numerically that all the single-particle states are strongly localized, and have only rare single-particle resonances, so that the model might be expected to be many-body localized (MBL) for sufficiently large $W$, by analogy with the standard MBL case[1]. We shall call case (b) symmetrized−MBL, as it obeys the symmetry embodied in Assumption 3.

We begin by verifying assumption 1 for both models, by numerically diagonalizing the Hamiltonian for systems sizes $L = 11 - 19$ setting $J = 2$ and $\Delta = 1$. We work in the zero (1/2) total magnetization sector for even (odd) system sizes. For the Stark-MBL case we take $\gamma = 2.75$ and for the symmetrized-MBL case $W = 9$. We use 10,000 disorder realizations for averaging. Figure 2 shows a histogram of the logarithm of the eigenvalue spacings, $\log_{10} \delta E$. Both models have a wide range of eigenvalue pairs that lie very close to each other compared to the average spacings, but are not degenerate. These quasi-degenerate pairs of states are found across the symmetry sectors of $\hat{P}$, as we show by restricting the eigenvalues to the even sector and calculating its distribution (see also Supplementary Note 2). The restricted distribution is centered around the average spacing and does not have a fat tail stretching to zero, which characterizes the unrestricted distribution. Since (8) satisfies all the assumptions of our proof, we expect that a finite fraction of its eigenstates are delocalized. To confirm this, we calculate $\overline{\sigma_\rho^2}$ in Eq. (5) within the microcanonical ensemble,

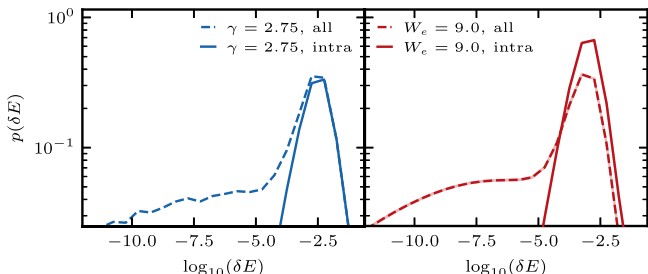

**Fig. 2 | (Color online) Distribution of level spacings $L = 18$ on a log–log scale for the Stark-MBL Hamiltonian, $\gamma = 2.75$ (left panel) and the symmetrized–MBL Hamiltonian, $W = 9$ (right panel).** Spacings restricted to the even parity symmetry sector (solid lines) and within the entire zero-magnetization sector (dashed lines). Statistical errors are denoted by line width.

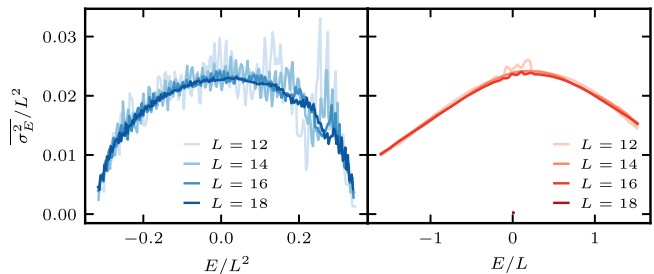

**Fig. 3 | (Color online) Rescaled infinite-time average of the microcanonical mean-square displacement $\overline{\sigma_E^2}/L^2$ in the zero-magnetization sector as a function of rescaled energy for system size $L = 12, 14, 16,$ and $18$ (darker shades correspond to larger systems).** The left panel corresponds to the Stark-MBL Hamiltonian with $\gamma = 2.75$ and the energy rescaled as $E/L^2$, and the right panel corresponds to the symmetrized−MBL Hamiltonian with $W = 9$, where the energy is rescaled as $E/L$.

$\hat{\rho}(E) = \mathcal{N}_E^{-1} \sum_{\alpha \in I} |\alpha\rangle \langle \alpha|$, where $I = [E - \Delta E, E + \Delta E]$ we take $\Delta E = \frac{\max(E) - \min(E)}{20}$, and $\mathcal{N}_E$ is the number of states in $I$. We also set $j = L/2$. Figure 3 shows that $\overline{\sigma_E^2}/L^2$ plotted vs rescaled energy for both models is nicely collapsed such that the states at all energies are delocalized, $\overline{\sigma_E^2} \sim L^2$.

We have established both analytically and numerically that both models have a delocalized excitation profile at all energy densities and infinite times. While this result is universal as long as Assumptions 1–3 are satisfied, the temporal and spatial dependence of the excitation profile (2) are model specific and are therefore left for the Supplementary Information. It is worthwhile to mention that both models exhibit subdiffusive transport which is better described by logarithmic subdiffusive transport, $t \sim \ln L$[43,44], and not power-law subdiffusive transport, $t - L^z$ $(z > 2)$[51,52], see Supplementary Note 5. For finite systems the spin-spin correlation function (2) decays to zero at all sites except $j$ and its mirror $\bar{j}$, which suggests a residual memory of initial conditions. The memory, however, fades away with increasing system size, see Supplementary Note 4.

## Symmetry breaking

The proof of finite spin transport crucially depends on the existence of the symmetry $\hat{P}$. It is interesting to see if finite transport persists also when the symmetry is broken. We have numerically examined a number of ways to break the symmetry in models described by (8): taking a finite magnetization, using an odd system size, or breaking the symmetry of the magnetic field $h_n$. All produce qualitatively similar behavior of dramatically suppressed dynamics (see for example Refs. 6,7). Here we only present results for odd system sizes and total magnetization 1/2. To examine the localization of the excitation profile (2), we compute a positive version of the MSD by taking $|G_{ij}^\rho(t) - G_{ij}^\rho(0)|$

in Eq. (4) and taking an infinite-time average, $\overline{\sigma^2_{\text{sgn}}}$. This is done to avoid the quasi-conservation of the MSD in Stark-MBL systems[53,54]. While it implies the absence of diffusion, it does not exclude subdiffusive transport[51,54]. Figure 4 shows that $\overline{\sigma^2_{\text{sgn}}}$ grows with system size for both models. It is hard to extract a reliable dependence on the system size from the accessible system sizes, but the growth is consistent with $L^{0.35}$ for the Stark-MBL system and $L^{1.35}$ for the symmetrized–MBL system. If the growth persists in the thermodynamic limit it implies asymptotic delocalization. Instead of breaking the symmetry of the Hamiltonian, we can use a nonequilibrium initial condition $\hat{\rho}$ which either satisfies or breaks the symmetry. For initial conditions that are odd or even with respect to $\hat{P}$ (such as the Néel state), we observe some memory of the initial state, however, there is no asymptotic memory retention of initial conditions that break the symmetry, see Supplementary Note 6.

## Discussion

We have proved that any spin chain with the symmetry given in Assumption 3 and a non-degenerate spectrum exhibits spin transport for a finite measure of its eigenstates in the zero-magnetization sector. The proof does not apply to noninteracting systems that have degeneracies due to the symmetry, and thus can remain localized. We have numerically demonstrated delocalization of the asymptotic excitation profile for two cornerstone models of localization in many-body systems: the Stark-MBL model with either purely linear potential or with any additional potential respecting $h_i = -h_{\bar{i}}$, and the symmetrized–MBL model. Our results suggest that for these models delocalization happens at all energy densities and spin transport is subdiffusive, and most probably logarithmic, see Supplementary Note 5. Moreover, our numerical results are consistent with asymptotic delocalization of the excitation profile also in the case of weak symmetry breaking, even though the dynamics is strongly suppressed for numerically accessible system sizes. Since some symmetry-breaking perturbation always exists in any experiment, and the time of running the experiment is inherently bounded, it might be extremely hard, if not impossible, to differentiate between glassy dynamics and bona fide localization.

### Constructing a localized delocalizing bath

A localized system is typically a closed system with no transport. Coupling a localized system to a Markovian heat bath induces slow transport for local coupling[55,56] or diffusive transport for global coupling[57–61]. A similar effect is expected to occur if a Markovian bath is replaced by a sufficiently large thermalizing system. But what if we couple two localized systems? Is it possible to induce transport in such a configuration? Since by definition there is transport in neither system, as a result of the coupling only resonant transfer between the two systems is possible. A possible guess could be coupling $\hat{H}$ to itself, that we will call the "$\hat{H}$ to $\hat{H}$" composite system. When the two systems are uncoupled all the spectrum is doubly degenerate and therefore

resonant. Under such conditions, any small coupling between the systems lifts the degeneracies and presumably results in weak transport. However numerical results suggest that this configuration does not result in delocalization for noninteracting systems, see Supplementary Note 7. We conclude that the existence of exact resonances is not a sufficient condition of delocalization.

We now use our results to construct a localized system which, when attached to the edge of a given localized system, described by a localized Hamiltonian $\hat{H}$, delocalizes it. Since $\hat{H}$ is localized, the unitarily transformed system $\hat{P}\hat{H}\hat{P}$ is also localized. For noninteracting systems the symmetry $\hat{P}$ implies that the single-particle spectrum of $\hat{P}\hat{H}\hat{P}$ is a reflection around zero of the single-particle spectrum of $\hat{H}$, such that there are no exact single-particle resonances, see Supplementary Note 1 (See Supplemental Information for a detailed proof and discussion of noninteracting systems and spatial and temporal dependence of the excitation profile). On the other hand, the many-body spectrum of $\hat{H}$ is identical to $\hat{P}\hat{H}\hat{P}$ and therefore has exact many-body resonances, similar to the $\hat{H}$ to $\hat{H}$ system. Nevertheless, by coupling $\hat{H}$ and $\hat{P}\hat{H}\hat{P}$ at the edge using a symmetric coupling $\hat{P}\hat{V}\hat{P} = \hat{V}$ results in a composite Hamiltonian that is symmetric under $\hat{P}$: $\hat{H}' = \hat{H} + \hat{P}\hat{H}\hat{P} + \hat{V}$ (see Fig. 1). We will call this coupling "$\hat{H}$ to $\hat{P}\hat{H}\hat{P}$". Since $\hat{H}'$ satisfies Assumptions 1–3 for a coupling $\hat{V}$ which breaks all degeneracies, it follows from the delocalization proof that $\hat{H}'$ is delocalized. The most dramatic demonstration of symmetry-induced delocalization can be obtained by coupling two Anderson insulators $\hat{H}$ and $\hat{P}\hat{H}\hat{P}$. A noninteracting coupling of the form $\hat{V} = \hat{S}^+_{L/2}\hat{S}^-_{L/2+1} + \hat{S}^-_{L/2}\hat{S}^+_{L/2+1}$ cannot lift the degeneracies and the system is localized. On the other hand, modifying $\hat{V}$ to include an interacting term, such as $\hat{S}^z_{L/2}\hat{S}^z_{L/2+1}$ lifts the degeneracies and results in delocalization via the delocalization proof. Thus, the $\hat{H}$ to $\hat{H}$ coupled system has resonances but appears to be localized, while the $\hat{H}$ to $\hat{P}\hat{H}\hat{P}$ coupling also has resonances, yet is delocalized. Studying the difference between these systems may provide insight into the role of resonances in delocalization.

### Outlook

In this work, we have considered the limit of infinite time for a finite system. For a finite time, the system might exhibit slow transport, see Supplementary Note 5, or appear to be localized[62]. It is an open and interesting question whether localization is possible also in the thermodynamic limit.

Other open questions are whether the delocalization mechanism carries over to other spatial symmetries. What is the fastest possible transport between two coupled localized systems? Is it always logarithmic? It would be also interesting to see if our delocalization proof can be generalized to higher dimensions, the microcanonical ensemble, other conserved quantities, such as the energy, and unbounded local Hilbert space dimensions. Moreover, implications on thermalization in systems respecting symmetry and the degree of stability of delocalization to symmetry-breaking perturbations should be also explored.

## Methods

Numerical data is obtained from a full eigendecomposition of the Hamiltonian in the appropriate symmetry sector in double floating point precision. We make use of the QUSPIN library[63] to generate Hamiltonians. For disordered systems, averaging is performed over 10,000 disorder realizations, and the standard deviation is estimated via the standard deviation of bootstrap samples. For small system sizes, the results have been compared to eigendecomposition with quadruple floating point precision via ADVANPIX[64].

## Data availability

All data is available upon reasonable request.

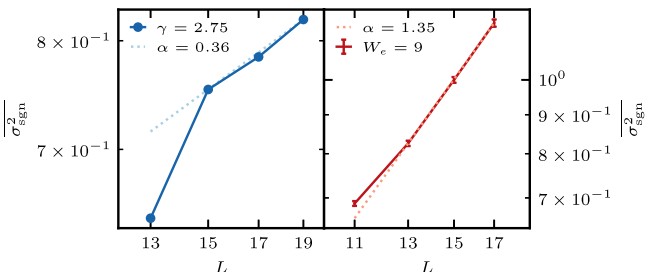

**Fig. 4 | (Color online) A log–log plot of $\overline{\sigma^2_{\text{sgn}}}$ as a function of $L$ for odd system sizes and total magnetization 1/2.** The left panel corresponds to the Stark-MBL Hamiltonian with $\gamma = 2.75$, and the right panel corresponds to the symmetrized-- MBL Hamiltonian with $W = 9$.

## Code availability

All code used in this work is available upon reasonable request.

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

## Acknowledgements

We thank B. Buča for comments on the manuscript. J.C.H. acknowledges funding from the European Research Council (ERC) under the European Union's Horizon 2020 research and innovation program (Grant Agreement no 948141)—ERC Starting Grant SimUcQuam, and by the Deutsche Forschungsgemeinschaft (DFG, German Research Foundation) under Germany's Excellence Strategy—EXC-2111—390814868. A.L. acknowledges support from EPSRC Grant No. EP/V012177/1. This research was supported by a grant from the United States-Israel Binational Foundation (BSF, Grant No. 2019644), Jerusalem, Israel, and by the Israel Science Foundation (grants No. 527/19 and 218/19). The Flatiron Institute is a division of the Simons Foundation.

## Author contributions

The project was initiated and led by Y.B. and A.L. Numerical experiments were performed by B.K. and J.C.H. and the proof was derived by Y.B. and B.K. Otherwise, all authors were involved equally at all stages of the project, in particular analysis and interpretation of the results as well as preparation of the manuscript.

## Competing interests

The authors declare no competing interests.
