## [Peer Review File · Nature Communications]

Reviewers' comments:

Reviewer #1 (Remarks to the Author):

The authors rigorously prove delocalization in strongly disordered spin chain models that conserve total magnetization and satisfy certain assumptions, namely, non-degeneracy of the many-body spectrum and a discrete symmetry under a combination of mirror and spin-flip. They support their calculations with numerics in well known models such as Stark MBL and a disordered spin chain where the disorder satisfies the symmetry assumption. Finally, they also show that two individually MBL systems satisfying the symmetry constraint can delocalize when brought in contact.

Their analytical proofs and supporting numerical data seem clear and are neatly presented. However, as the author's begin discussing in their introduction, there have been a number of studies exploring the role symmetries play in the physics of MBL and in particular, settings where MBL is impossible. One such example is models with non-Abelian continuous symmetries that do not many body localize. Also, there is now increasing evidence that questions the existence of truly MBL physics in well known models which the authors also discuss.

This brings me to the conclusion that the findings in the paper do not seem to significantly advance understanding MBL physics, but rather, only an addition to the models that do not MBL.. This holds me back in recommending publication in Nature Communications. Having said that, since the authors also make interesting points about transport in these models and consequences with violating symmetry assumptions, it would be quite useful to throw more light on them and proposals for experiments could make a significant impact and qualify for publication.

Reviewer #2 (Remarks to the Author):

Much attention in recent years has been focussed on the relaxation of many-body quantum systems. In contrast to expectation that interacting quantum systems should quickly thermalise and are ergodic, many settings of various robustness have been found that exhibit non-ergodicity. Many-body localisation in particular, has been shown to be generically robust to perturbations in 1D, and provides a way of avoiding thermalising completely. If we also consider transient or finite time dynamics of quantum systems, then the situation is even richer, where there are a number of ways

of obtaining slow dynamics, or atypical infinite temperature eigenstates. This work adds to this rich story by showing that discrete symmetries can prevent localisation.

The results in this work are to my knowledge novel and of fundamental interest to the field of non-equilibrium quantum dynamics. The results seem sound and the authors offer up several interesting questions in their outlook. This paper certainly deserves publication, and is well suited to this journal.

Before publication, however, I have one conceptual question that I would like to be addressed:

- one of the examples of where your delocalisation argument applies is to the stark-MBL system. However, unless I have made a mistake, it seems that the model you consider is the same as the original proposal up to constants (if we are in a fixed magnetization sector). Therefore, your work seems to be in disagreement with the original stark-MBL results. Please can you explain?

I also think some minor improvements to the presentation can be made:

- The introduction currently reads a collection of paragraphs on different things. While they are all related, there is no clear narrative, which makes it more difficult to read and to ascertain the main result of the paper and its context. Paragraph 4 is particularly cryptic and relies heavily on jargon and pointing to reference.

- I think there is an error in Eq.(2), I think the second part should also have a time dependence.

- Non-degeneracy is used to get to Eq.(3). However, I would be interested to know more about how the arguments break down when there is a degenerate spectrum. This is commented on throughout, but if it's easy to show where the logic breaks down and what degeneracies would allow that would be more convincing in my opinion.

- Eq.(4) appears to have an implicit dependence on j . This is in principle fine, but the quantity plotting in fig.3 does not have this dependence, despite the same notation. Is the value in Fig.3 averaged over j as well? Please clarify in the text.

- Minor note: a tilde is used in the main text to indicate mirrored coordinates, whereas a prime is used in the Sup.Mat.

- From reading the Sup.Mat. Eq.(6) requires the symmetry P (as stated), but also that we are in the zero magnetization sector. While this is one of your assumptions for the proof, it is not clear as written that both are required for Eq.(6).

- I am fine with the "proof", however, one might argue that is not a proof, simply because there is not a clear statement of what you are proving. To get closer to a proof, you could prove that the infinite-time MSD scales as L^2 , and then the physical interpretation would be that the system is delocalized.

- The last two paragraphs before the discussion refer to results that are not included in the main text. I think the way it is currently written is a bit confusing. It would be nicer to clearly say you also tested other things, this is shown in the Sup.Mat., the results were X, Y, Z, and this is consistent with everything else we write in the main text.

Once my comments have been addressed, I would be happy to recommend this work for publication.

Reviewer #3 (Remarks to the Author):

The aim of the paper is to prove that a finite fraction of the eigenstates within the zero magnetization sector of systems with a particular symmetry are delocalized. This is done by determining the scaling with system size of the mean-squared displacement defined in equation (4). Two models are studied further numerically to demonstrate the delocalization.

I do not think this paper will have a big impact on the field. The idea that symmetry can result in delocalization is not new. See, e.g., references [29] and [30], and the symmetry considered here seems rather specialized. It is not clear how it could arise naturally. The claims made in the paper seem slightly exaggerated (see comments below), and the presentation could be improved.

Comments:

- The Authors claim that the proof relies only on assumptions 1-3. This is misleading, as they also assume the total magnetization is zero in the proof. The claim about delocalization therefore only

applies if the zero-magnetization sector is populated. One could have a system that fulfils assumptions 1-3, but if only, e.g., the sector with total magnetization plus/minus 1 is populated, this system does not need to be delocalized. It should also be mentioned in the section about assumptions that the proof relies on taking the infinite time average for a finite system size.

- The sentence "We have proved that any spin chain with the symmetry given in Assumption 3 and a nondegenerate spectrum exhibits spin transport for a finite measure of its eigenstates." is incorrect. Assumption 2 and a comment about the zero magnetization sector should be added.

- In the abstract the authors write "and show that the delocalization mechanism is robust to breaking the symmetry". It should be made clear what exactly is shown. If one takes a finite system and deviates sufficiently little from the symmetric point, one would expect to see the same physics as for the symmetric case. The relevant information is hence how much one can deviate before different physics appears. The Authors should also make clear that this statement is based on a numerical observation for a particular system.

- Between (S13) and (S14) the authors write "Since the total magnetization is set to zero, the total spin $\sum_i \vec{S}_i$ is rotationally invariant". That $\sum_i S_i^z = 0$ is not by itself enough to ensure that the total spin is rotationally invariant. For instance, the $S^z=0$ component of a spin-1 is not rotationally invariant. A different argument is needed to justify (S14).

- The claim "proves that the existence of exact resonances is not a sufficient condition for delocalization" in the abstract seems to rely on results that are not shown, so it is unclear how much substance there is in this.

Minor points:

- Page 1, left column: "Lowering the disorder strength increases the density of the resonances and eventually leads to delocalization at $d \geq 3$ [12]." What is d ?

- (S1) and (S3) only agree if f is real, but this has not been stated.

- Equation (S3): The last index k should be an i .

- Equation (S30): The last sum should not be there.

- There are several typos that should be corrected.

Reviewer #1 (Remarks to the Author):

The authors rigorously prove delocalization in strongly disordered spin chain models that conserve total magnetization and satisfy certain assumptions, namely, non-degeneracy of the many-body spectrum and a discrete symmetry under a combination of mirror and spin-flip . They support their calculations with numerics in well known models such as Stark MBL and a disordered spin chain where the disorder satisfies the symmetry assumption. Finally, they also show that two individually MBL systems satisfying the symmetry constraint can delocalize when brought in contact.

Their analytical proofs and supporting numerical data seem clear and are neatly presented. However, as the author's begin discussing in their introduction, there have been a number of studies exploring the role symmetries play in the physics of MBL and in particular, settings where MBL is impossible. One such example is models with non-Abelian continuous symmetries that do not many body localize. Also, there is now increasing evidence that questions the existence of truly MBL physics in well known models which the authors also discuss.

This brings me to the conclusion that the findings in the paper do not seem to significantly advance understanding MBL physics, but rather, only an addition to the models that do not MBL.. This holds me back in recommending publication in Nature Communications.

We thank the Referee for the positive evaluation of our proof and the numerical results. We want to point out that while the role of symmetry was discussed in previous works, *our work constitutes the first rigorous proof of delocalization due to a discrete symmetry.* Before our work, localization was believed to be possible in discrete Abelian symmetries [29].

In the introduction, we mention that the existence of MBL, per se, without any symmetries is under debate [33-35]. This highlights the importance of rigorous studies , which resolve this argument which is difficult to conclusively resolve using approximate techniques. Our study proves that a broad family of systems cannot localize, and we numerically show that our proof is stable to certain symmetry-breaking perturbations. This can pave the way to delocalization proofs in more general settings.

Having said that, since the authors also make interesting points about transport in these models and consequences with violating symmetry assumptions, it would be quite useful to throw more light on them and proposals for experiments could make a significant impact and qualify for publication.

Our study suggests that it might be impossible to determine in experiments whether a system is genuinely localized or have glassy features since delocalization might occur at exceedingly long times. We have clarified this point in the text.

Reviewer #2 (Remarks to the Author):

Much attention in recent years has been focussed on the relaxation of many-body quantum systems. In contrast to expectation that interacting quantum systems should quickly thermalise and are ergodic, many settings of various robustness have been found that exhibit non-ergodicity. Many-body localisation in particular, has been shown to be generically robust to perturbations in 1D, and provides a way of avoiding thermalising completely. If we also consider transient or finite time dynamics of quantum systems, then the situation is even richer, where there are a number of ways of obtaining slow dynamics, or atypical infinite temperature eigenstates. This work adds to this rich story by showing that discrete symmetries can prevent localisation.

The results in this work are to my knowledge novel and of fundamental interest to the field of non-equilibrium quantum dynamics. The results seem sound and the authors offer up several interesting questions in their outlook. This paper certainly deserves publication, and is well suited to this journal.

Before publication, however, I have one conceptual question that I would like to be addressed:

- one of the examples of where your delocalisation argument applies is to the stark-MBL system. However, unless I have made a mistake, it seems that the model you consider is the same as the original proposal up to constants (if we are in a fixed magnetization sector). Therefore, your work seems to be in disagreement with the original stark-MBL results. Please can you explain?

We thank the Referee for the positive evaluation of our work. Almost all Stark-MBL results are studied in the presence of a weak symmetry-breaking perturbation (for example, disorder or a weak parabolic potential that's scaled down with the system size to avoid localization due to the confining potential). Before our work, the existence of localization in the purely linear system was under debate, with different works concluding or implying that it is or not stable. Our work conclusively resolves this question.

I also think some minor improvements to the presentation can be made:

- The introduction currently reads a collection of paragraphs on different things. While they are all related, there is no clear narrative, which makes it more difficult to read and to ascertain the main result of the paper and its context. Paragraph 4 is particularly cryptic and relies heavily on jargon and pointing to reference.

We have amended the introduction and specifically paragraph 4, to make it more cohesive.

- I think there is an error in Eq.(2), I think the second part should also have a time dependence.

The equation is actually correct in the context it is stated in, since for the density matrix which is an equilibrium state, $\langle Sz(t) \rangle = \langle Sz \rangle$.

- Non-degeneracy is used to get to Eq.(3). However, I would be interested to know more about how the arguments break down when there is a degenerate spectrum. This is commented on throughout, but if it's easy to show where the logic breaks down and what degeneracies would allow that would be more convincing in my opinion.

When there are degeneracies then Eq(3) will have off-diagonal elements of the form, $\langle \alpha | Sz | \beta \rangle$, and a summation over all the multiplicity of E_{α} . Currently, we don't know how to derive an equivalent of Eq(6) for terms of the form $\langle \alpha | Sz | \beta \rangle$.

- Eq.(4) appears to have an implicit dependence on j . This is in principle fine, but the quantity plotting in fig.3 does not have this dependence, despite the same notation. Is the value in Fig.3 averaged over j as well? Please clarify in the text.

In the numerical calculation we set $j=L/2$. We have added this missing detail to the text.

- Minor note: a tilde is used in the main text to indicate mirrored coordinates, whereas a prime is used in the Sup.Mat.

Thank you, fixed.

- From reading the Sup.Mat. Eq.(6) requires the symmetry P (as stated), but also that we are in the zero magnetization sector. While this is one of your assumptions for the proof, it is not clear as written that both are required for Eq.(6).

For nonzero magnetization Assumptions 2 and 3 are inconsistent, therefore a system which satisfies both assumptions must be in the zero magnetization sector. This is discussed just below the assumptions.

- I am fine with the "proof", however, one might argue that is not a proof, simply because there is not a clear statement of what you are proving. To get closer to a

proof, you could prove that the infinite-time MSD scales as L^2 , and then the physical interpretation would be that the system is delocalized.

Since Nature Communications is a general audience journal we have decided against formal mathematical style. The objective of the proof is therefore stated not within a Theorem clause but in a paragraph just below Eq(4). While the presentation is not formal, the proof is rigorous.

- The last two paragraphs before the discussion refer to results that are not included in the main text. I think the way it is currently written is a bit confusing. It would be nicer to clearly say you also tested other things, this is shown in the Sup.Mat., the results were X, Y, Z, and this is consistent with everything else we write in the main text.

In the mentioned paragraphs we cite the Supplementary (Ref [52]). Moreover, the contents of the supplementary is detailed in the reference list.

Once my comments have been addressed, I would be happy to recommend this work for publication.

Reviewer #3 (Remarks to the Author):

The aim of the paper is to prove that a finite fraction of the eigenstates within the zero magnetization sector of systems with a particular symmetry are delocalized. This is done by determining the scaling with system size of the mean-squared displacement defined in equation (4). Two models are studied further numerically to demonstrate the delocalization.

I do not think this paper will have a big impact on the field. The idea that symmetry can result in delocalization is not new. See, e.g., references [29] and [30], and the symmetry considered here seems rather specialized. It is not clear how it could arise naturally. The claims made in the paper seem slightly exaggerated (see comments below), and the presentation could be improved.

We thank the Referee for his/her assessment. Let us begin by pointing out that there is a difference between ideas and the ability to prove or justify them. Any proof is important because it provides a solid base for future directions especially if, like in the case of MBL, the actual existence of something is under dispute. On the other hand, a novel idea might be incorrect, no matter how exciting.

To our knowledge, there are no known examples of *discrete* symmetries that lead to the delocalization of a finite measure of states. The works that the Referee mentions in

his/her report, and we cite in the introduction ([29] and [30]), rule out MBL in *continuous non-Abelian* symmetries. Moreover, these works are not-rigorous and assume a specific localization structure (the I-bit picture). Our work is therefore the first, *rigorous*, example of delocalization in a system with discrete symmetry.

We are surprised by the Referee's comment, "the symmetry is rather specialized". By definition, any symmetry can be called special. On the other hand, any physical system feeling a spatially constant force (such as gravity and dc electric field) satisfies this symmetry. Moreover, cold atoms experiments on Stark-MBL were performed (see Refs. [10-11]) so that this symmetry class includes several very important realistic potentials, which is what motivated our study in the first place.

Comments:

- The Authors claim that the proof relies only on assumptions 1-3. This is misleading, as they also assume the total magnetization is zero in the proof. The claim about delocalization therefore only applies if the zero-magnetization sector is populated. One could have a system that fulfils assumptions 1-3, but if only, e.g., the sector with total magnetization plus/minus 1 is populated, this system does not need to be delocalized.

Assumptions 2 and 3 are inconsistent unless the total magnetization is zero. Therefore any system satisfying Assumptions 1-3 must be with total magnetization. It is *not* an additional assumption. This point is clearly stated just below the assumptions in the main text.

It should also be mentioned in the section about assumptions that the proof relies on taking the infinite time average for a finite system size.

In the sentence leading to Assumption 1, it is stated that we consider a one-dimensional chain of length L . The fact that the infinite-time average is used is not an assumption but a technical tool we use for the proof. It only relies on Assumption 1.

- The sentence "We have proved that any spin chain with the symmetry given in Assumption 3 and a nondegenerate spectrum exhibits spin transport for a finite measure of its eigenstates." is incorrect. Assumption 2 and a comment about the zero magnetization sector should be added.

We disagree with the Referee that the statement is incorrect since the existence of spin transport implies conservation of magnetization, which is precisely Assumption 2. As stated in our reply above, to be consistent with Assumption 3, this implies zero magnetization. Notwithstanding, to be more explicit, we highlighted that the proof applies only at zero magnetization.

- In the abstract the authors write "and show that the delocalization mechanism is robust to breaking the symmetry". It should be made clear what exactly is shown. If one takes a finite system and deviates sufficiently little from the symmetric point, one would expect to see the same physics as for the symmetric case. The relevant information is hence how much one can deviate before different physics appears. The Authors should also make clear that this statement is based on a numerical observation for a particular system.

We have clarified the abstract in this respect. The primary purpose of our work was to prove delocalization. Since it is not evident that the limit of a symmetric system is not singular, different physics might appear for any nonzero symmetry-breaking interaction. Our numerical simulations point out that this is not the case.

- Between (S13) and (S14) the authors write "Since the total magnetization is set to zero, the total spin $\sum_i \vec{S}_i$ is rotationally invariant". That $\sum_i S_i^z = 0$ is not by itself enough to ensure that the total spin is rotationally invariant. For instance, the $S^z=0$ component of a spin-1 is not rotationally invariant. A different argument is needed to justify (S14).

We agree with the Referee that the sentence before (S14) was incorrect, and the result we obtained in (S16) is only correct asymptotically. Since the exact expression for $\langle (S^z_i)^2 \rangle$ is *not* essential for the rest of the proof, we set it as $\langle (S^z_i)^2 \rangle = C_s$, where C_s is a constant of the order of $O(s^2)$. By equivalence of ensembles, in the limit of $L \rightarrow \infty$, C_s goes to (S16). So (S16) is asymptotically exact. Moreover, for $s=1/2$, (S16) is correct for any system size. We have fixed those points in the Supplementary.

- The claim "proves that the existence of exact resonances is not a sufficient condition for delocalization" in the abstract seems to rely on results that are not shown, so it is unclear how much substance there is in this.

In the discussion, we have provided an example of a system (Anderson insulator with a symmetric potential) with exact resonances but which is nevertheless localized. The localization was not proven in the text (although we suspect that standard proofs of Anderson localization still apply here) but was numerically verified (not shown in the previous version). We have added numerical evidence to this localization in the revised version of the Supplementary material. Therefore, we have shown that exact resonances are not sufficient for delocalization as we hope the referee will now agree.

Minor points:

- Page 1, left column: "Lowering the disorder strength increases the density of the resonances and eventually leads to delocalization at $d \geq 3$ [12]." What is d ?

Dimension. We have fixed this in the text.

- (S1) and (S3) only agree if f is real, but this has not been stated.

It follows from the symmetry that f is real. We have stated this explicitly.

- Equation (S3): The last index k should be an i .

Thanks. Fixed.

- Equation (S30): The last sum should not be there.

Of course. Thanks.

- There are several typos that should be corrected.

We proofread the text once more.

REVIEWERS' COMMENTS

Reviewer #1 (Remarks to the Author):

The authors have justly answered my comments. I would be happy to recommend this paper for publication.

Reviewer #3 (Remarks to the Author):

The reply is not convincing. I still think this work is too specialized for publication in Nature Communications, and I recommend rejection.

The considered symmetry is highly specialized, because physical parameters for sites that are far apart are required to take exactly the same value except for a sign, and this is not a physically natural constraint. For the disordered model in equation (8), e.g., the magnetic fields at site n and $L+1-n$ are required to be exactly the same, except for a sign, while the magnetic field is also assumed to vary randomly from one site to the next. The assumption of zero magnetization further narrows the results.

As mentioned in my previous report, and also by the first referee, studying the effect of symmetry on localization is not new. With reference to [29] and [30], the authors write "these works are not-rigorous and assume a specific localization structure (the I-bit picture)". In other words, these works rely on particular assumptions, but the present work also relies on assumptions, and these assumption are rather specialized. In addition, the investigations of the effects of breaking the symmetry are not sufficiently solid, and the statements made about the symmetry breaking do not sufficiently clearly state what those statements are based on.

Altogether, I do hence not think this work will have a big impact on the field.

Dear Editor,

We thank you and the Reviewers for the review of our manuscript.

Reviewer 3

“The reply is not convincing. I still think this work is too specialized for publication in Nature Communications, and I recommend rejection. The considered symmetry is highly specialized, because physical parameters for sites that are far apart are required to take exactly the same value except for a sign, and this is not a physically natural constraint. For the disordered model in equation (8), e.g., the magnetic fields at site n and $L+1-n$ are required to be exactly the same, except for a sign, while the magnetic field is also assumed to vary randomly from one site to the next. The assumption of zero magnetization further narrows the results.”

Perhaps putting our work in the context of current research will help here. There has been a very significant number of publications, both theoretical and experimental, studying so-called Stark MBL. This is an interacting system with a linear potential, which is a special case of what we study. The reason this model has attracted so much attention is because even without disorder the system is localised in the non interacting limit.

Our work rigorously shows that a broad class of systems, including this very well-studied one, is *not* localised. Previous work has relied on numerics, and has not led to a consensus.

We have therefore *conclusively* shown that a model very intensively studied in the hope of it being localised is not, in fact, localised. Our result is, however, much more general. As a way of emphasising just how surprising the result is, we provided an additional example of a potential satisfying this symmetry that intuitively might be expected to be localised, namely, an antisymmetric but otherwise disordered potential.

In conclusion: Yes, the antisymmetric disordered potential has long-range correlations. But the value of this work doesn't lie in studying this specific potential, but rather in that it shows that a broad class of potentials, including a very well-studied, non-disordered one that has been introduced and widely studied in the hope that it is localised, is unambiguously not localised.

Even ignoring the importance of this model, such exact results are rare in the literature on localisation.

As mentioned in my previous report, and also by the first referee, studying the effect of symmetry on localization is not new. With reference to [29] and [30], the authors write "these works are not-rigorous and assume a specific localization structure (the 1-bit picture)". In other words, these works rely on particular assumptions, but the present work also relies on assumptions, and these assumption are rather specialized.”

We have proven delocalisation for an entire class of models satisfying certain criteria. This set of criteria was selected so that a special case is a widely-studied potential, causing localisation in the noninteracting case and widely studied in the hope that it also showed localisation. That is, it is not some special case we arbitrarily decided to study.

“In addition, the investigations of the effects of breaking the symmetry are not sufficiently solid, and the statements made about the symmetry breaking do not sufficiently clearly state what those statements are based on.”

This concern has already been addressed in the revised version of our manuscript, see the response to a similar concern of Referee 3 in responses submitted with the first revised version. We discuss clearly in the results section what kind of symmetry breaking we are considering (finite magnetization) and in which sense we consider our numerical observations indicative of asymptotic delocalization under the presence of weak, but finite symmetry breaking. In addition, we phrase the relevant statement in the conclusion explicitly in reference to our numerical results. We also state that a more careful study of the influence of symmetry breaking is an important future direction.

“Altogether, I do hence not think this work will have a big impact on the field.”

We respectfully disagree.